# Analysis of Serum Th2 Cytokines in Infants with Non-IgE Mediated Food Allergy Compared to Healthy Infants

**DOI:** 10.3390/nu14081565

**Published:** 2022-04-09

**Authors:** Francesco Savino, Francesca Giuliani, Stefano Giraudi, Ilaria Galliano, Paola Montanari, Valentina Daprà, Massimiliano Bergallo

**Affiliations:** 1Department of Pediatrics, Regina Margherita Children Hospital, Città della Salute e della Scienza di Torino, Piazza Polonia 94, 10126 Torino, Italy; giuliani.pediatria@gmail.com; 2Postgraduate School of Pediatrics, University of Turin, Piazza Polonia 94, 10126 Torino, Italy; stefano.giraudi95@gmail.com; 3Department of Public Health and Pediatric Sciences, University of Turin, 10126 Torino, Italy; ilaria.galliano@unito.it (I.G.); paola.montanari@unito.it (P.M.); valentina.dapr@yahoo.it (V.D.); massimiliano.bergallo@unito.it (M.B.)

**Keywords:** infants, non-IgE food allergy, interleukins, hydrolyzed formula

## Abstract

Background: The aim of this study is to assess the serum values of IL-4, IL-5, IL-10, and IL-13 in a group of infants with non-IgE mediated food allergies treated with a hydrolyzed formula and compare them with a group of healthy peers. Methods: A total of 53 infants aged 1 to 4 months, of which 34 with non-IgE mediated food allergies and 19 healthy infants were enrolled in this study. Infants were eligible if they had gastrointestinal symptoms of food allergy and needed to switch from their initial formula to hydrolyzed formulas with an improvement of symptoms. Controls were fed with either breastmilk or standard formula. Blood samples were taken within one week of a special diet for cases. Interleukinsin in peripheral blood was detected and analyzed using the real-time PCR MAMA method. Fecal calprotectin was evaluated using a quantitative assay. Results: Values of IL-4 and IL-13 were significantly higher in the non-IgE food allergy group compared to the control group (*p* < 0.05), while IL-5 and IL-10 were significantly lower than the control group (*p* < 0.05). Fecal calprotectin in the non-IgE food allergy group was significantly higher compared to the control group (*p* < 0.05). Conclusion: This study provides a theoretical basis that Th2 cytokine expression in infants with a non-IgE mediated food allergy is significantly different than in healthy infants; this finding supports the use of early dietetic treatment with hydrolyzed formulas.

## 1. Introduction

Non-IgE-mediated food allergies are immunologic reactions to food that present without serum-specific IgE antibodies and may occur in early life [1].

The new guidelines of WHO for cow’s milk allergy diagnosis and treatment include the entity of non-IgE mediated food allergy in the most recent update, thus recognizing the relevance of this condition both in adults and particularly in children [2].

From the immunological point of view, in the most frequent forms of CMPA, an abnormal proliferation of the Th2 subclass of T-lymphocytes occur and determines the production of lymphokines—such as IL-4—which induce an abnormal production of IgE by B-lymphocytes and amplify the inflammatory response and the production of Th2-associated cytokines (IL-5, IL-10, and IL-13), involved in atopic reactions [3].

Atopy, in general, is a predisposition to respond immunologically to allergens leading to Th2 differentiation and induction of immunoregulatory mechanisms that occur via some cytokines production, such as IL-4, IL-5, IL-10, and IL-13 [4].

Interleukins are small signaling proteins that have central roles in inflammation in subjects with cow’s milk allergy and may lead to a deeper knowledge of the pathogenesis of food allergy [5].

In fact, despite advances in research, the pathophysiology of non-IgE-mediated food allergy has not been fully elucidated yet and it seems that the complex pathways leading to Th2 immune responses against food antigens cannot be attributed to a single driving force [6].

While the immediate symptoms of immunoglobulin E (IgE) mediated CMA are readily recognized, the diagnosis of non-IgE mediated CMA can pose a challenge due to the delayed onset of symptoms and overlap with other common pediatric manifestations of functional gastrointestinal disorders such as infantile colic, gastroesophageal reflux (disease) and even infections, particularly in the first months of life [1,2,6].

Non-IgE-mediated gastrointestinal food allergies are a heterogeneous group of food allergies in which there is an immune reaction against food, but the primary pathogenesis is not a production of IgE and activation of mast cells: therefore, the pathway of interleukins response needs to be clarified [1].

The major reason for the limited understanding of non-IgE food allergy is a delayed and less severe reaction to food ingestion and the fact that patients’ symptoms often improve with empiric food avoidance. Therefore, endoscopy and biopsy are not performed [6].

As a consequence, if clinical symptoms indicate a non-IgE-mediated CMA, it is recommended to propose a diet with the elimination of cow’s milk proteins and subsequent reintroduction: extensively hydrolyzed formula (eHF) whey or casein-based are considered the first-line management of formula-fed infants with CMPA and suggested as therapy [6,7,8,9].

Subjects with non-IgE-mediated food allergy were proposed to be fed with an extensively hydrolyzed formula, leading to improvement of symptoms compared to what happens in atopic infants; however, laboratory data on their cytokines response are scanty [6].

The importance to feed these subjects a diet similar to that of atopic has recently been suggested in the literature, observing an overall clinical improvement that is still not supported by laboratory evidence [9].

The aim of this study was to evaluate the Th2 cytokine response (IL-4, IL-5, IL-10, IL-13) in a group of infants with non-IgE-mediated food allergic symptoms and compare it to the response in healthy peer control.

Fecal calprotectin, which is a marker of gut inflammation whose use is emerging also in gastrointestinal allergic disorders [10,11], has been measured in both groups.

## 2. Materials and Methods

### 2.1. Subjects

#### 2.1.1. Infants

We screened for enrollment of all infants who were admitted to the Outpatient Clinic of the Special Care Unit of the Department of Pediatrics of the University of Turin, Regina Margherita Children’s Hospital, between October 2018 and October 2020. After discharge, the infants underwent blood tests during routine outpatient examinations. The study protocol was approved by the local Ethical Committee at Ospedale Mauriziano—Ospedale Infantile Regina Margherita—S. Anna Torino and written consent was obtained from infants’ parents.

Criteria for enrollment were as follows:Group 1: patients with mild to moderate non-IgE-mediated CMA. Patients within 4 months of life, fed with adapted formula milk, with gastrointestinal (abdominal discomfort, colic and fussing, food refusal, vomiting, diarrhea, constipation, hematochezia) and/or cutaneous symptoms (skin rash, itching, erythema, atopic dermatitis), and negative serum IgE to cow milk.Group 2: healthy infants of comparable age and not following any special diet, enrolled during outpatient routine health checks.

We considered eligible for the study patients with mild to moderate non-IgE- mediated CMA that needed to switch from their initial formula to special formulas (extensively casein hydrolyzed formulas or amino acid-based formulas) with an improvement of symptoms. Controls were fed with either breastmilk or standard formula. In the group of cases, blood samples were taken within one week from the special diet introduction. Venous blood sampling was taken from 7.30 to 8.30 a.m. during routine clinical.

#### 2.1.2. Serology

Food-specific IgE for casein was determined by processing patients’ serums from venous blood samples obtained at the time of enrollment with the automated Pharmacia CAP system RAST (Pharmacia & Upjohn Diagnostics, Uppsala, Sweden), and the cut-off point for positivity was set at 0.35 kUA/L.

#### 2.1.3. Fecal Calprotectin

Feces samples (about 5–10 g) were collected from each infant directly from the diaper and instantly analyzed for calprotectin values.

For the determination of fecal calprotectin value, a rapid test was employed, the BÜHLMANN Quantum Blue^®^ Calprotectin High Range (Schönenbuch, Switzerland). The analysis mechanism is based on a sandwich immunoassay with a measurement range between 100 and 1800 µg/g; it is for a quantitative assessment of fecal calprotectin value.

### 2.2. Total RNA Extraction

The automated extractor Maxwell (Promega, Madison, WI, USA) following the RNA Blood Kit protocol without modification was used to extract total RNA from whole blood samples. This equipment works with treatment with DNase during the RNA extraction process. Traditional Ultraviolet (UV) spectroscopy with absorbance at 260 and 280 nm was used to determine both purity and concentration of RNA. 

The Beer-Lambert law, which can describe a linear change in absorbance with concentration, allowed us to calculate the nucleic acid concentration. The RNA concentration range was in accordance with the manufacturer terms for the NanoDrop (Thermo Fisher Scientific, Waltham, MA, USA). UV absorbance measurements were acquired using 1 µL of RNA sample in an ND-1000 spectrophotometer under the RNA-40 settings at room temperature (RT). Applying this equation, an A260 reading of 1.0 is equivalent to ~40 µg/mL of single-stranded RNA. RNA purity was defined as the A260/A280 ratio, considering an A260/A280 ratio of 1.8/2.1 indicative of an RNA highly purified. RNA extracts were directly amplified without reverse transcription to verify the genomic DNA contamination. The RNAs were put in storage at −80° until testing.

### 2.3. Reverse Transcription

An amount of 400 nanograms of total RNA was reverse-transcribed with 2 μL of buffer 10×, 4.8 μL of MgCl2 25 mM, 2 μL ImpromII (Promega), 1 μL of RNase inhibitor 20U/l, 0.4 μL random hexamers 250 μM (Promega), 2 μL mix dNTPs 100 mM (Promega), and dd-water to obtain a final volume of 20 μL. The reaction mix was analyzed in a GeneAmp PCR system 9700 Thermal Cycle (Applied Biosystems, Foster City, CA, USA) under the following conditions: 5 min at 25 °C, 60 min at 42 °C, and 15 min at 70 °C for the inactivation of the enzyme. The cDNAs were kept at −20° until testing.

### 2.4. Transcription Levels of IL4, IL5, IL10, and IL13 by Real-Time PCR Assays

We chose Glyceraldhyde triphosphate dehydrogenase (GAPDH) as the reference gene in all determinations, because it is known to be the most stable among reference genes and has already been validated in our previous study [12]. The ABI PRISM 7500 real-time system (Thermo Fisher Scientific) allowed us to obtain the relative quantification of mRNA concentrations of IL4, IL5, IL10, and IL13.

We assessed the expression values of IFN-1, IFN-2, and IFN-3 using 40 ng of cDNA in a 20 μL of total volume reaction containing 2.5 U goTaQ MaterMix (Promega), 1.25 mmol/L MgCl2, 500 nmol of specific primers and 200 nmol of specific probes as reported in Table 1. Probes designed by Primer ExpressTM software version 3.0 (Thermo Fisher Scientific) were used. Analysis was conducted in a 96-well plate at 95 °C for 10 min, followed by 45 cycles at 95 °C for 15 s and at 60 °C for 1 min to run the amplification. Each sample was run in triplicate. Δelta-Δelta-Ct Algorithm was employed for relative quantification of target gene transcripts. Therefore, fold change was obtained and expressed in corresponding arbitrary units, named Relative Quantification (RQ). Since we measured Ct for every target in all samples, we applied the above-described methods for IL4, IL5, IL10, and IL13 detection and quantification [12].

### 2.5. Statistical Analysis

The data were analyzed with GraphPad Prism 7.01^®^. The quantitative variables were described in terms of medians (Me) and variation ranges. The qualitative variables were described through absolute frequencies and percentages. Depending on their distribution, comparisons of quantitative data samples were analyzed with Fisher’s test or the Mann-Whitney test. All tests were two-tailed and considered significant at values of *p* < 0.05.

Mann-Whitney test was used to compare the transcriptional levels of each IL between each group of children with each other. Statistical analyses were done using the Prism software (GraphPad Software, La Jolla, CA, USA). In all analyses, *p* < 0.05 was taken to be statistically significant.

## 3. Results

### 3.1. Subjects Characteristic

We enrolled a total of 53 subjects. Of these, 34 infants had symptoms of non-IgE-mediated food allergy and 19 were healthy controls.

IgE determinations in order to identify sensitization were performed and were found to be negative for both groups of infants.

We defined “non-IgE-mediated food allergic patients” as those who had gastrointestinal symptoms (diarrhea, vomiting, hematochezia, constipation), and/or cutaneous (eczema) and negative serum IgE to casein and whey protein and were then treated with hydrolyzed cow’s milk formula [6].

The median postnatal age at enrollment was 10 weeks in patients with mild to moderate non- IgE-mediated CMA and 8 weeks in healthy controls. In the group with mild to moderate non-IgE-mediated CMA 18 (53%) subjects were male, in the healthy group 9 (47.4%).

The median birth weight was 3180 g in the patients with mild to moderate non-IgE-mediated CMA and 3070 g in the healthy subjects. The median gestational age was 36 weeks in the group with mild to moderate non-IgE-mediated CMA and 37 in healthy infants, making the two groups reliably comparable.

The patients with mild to moderate non-IgE-mediated CMA25 were Italian (73%), and 14 (74%) were in the healthy group.

Feeding type: 29 patients (85%) with mild to moderate non-IgE-mediated CMA were fed with hydrolyzed casein formula and 5 (15%) with amino acid-based formula. Twelve healthy infants were exclusively breastfed (63%) and 7 formula-fed (37%).

Table 2 describes baseline characteristics, Table 3 describes Fecal calprotectin value.

### 3.2. Transcriptional Level of IL4, IL5, IL10, and IL13

As detailed in Figure 1, the expression values of IL4, IL5, IL10, and IL13 were significantly different in patients with mild to moderate non-IgE-mediated CMA patients versushealthy control.

In particular, the IL4 values (mean +/− SD) were higher: 1.33 +/− 0.97 in patients with mild to moderate non-IgE-mediated CMA topic patients vs. 0.80 +/− 1.11 in healthy controls, *p* = 0.006; the IL5 values were lower: 0.70 +/− 0.48 inpatients with mild to moderate non-IgE-mediated CMA vs. 1.11 +/− 0.99 in healthy controls *p* = 0.0294; the IL10 values were lower: 0.73 +/− 1.12 in patients with mild to moderate non-IgE-mediated CMA atopic patients vs. 1.05 +/− 1.54 in healthy control, *p* = 0.0198, and the values of IL13 were higher1.95+/− 0.77 in patients with mild to moderate non-IgE-mediated CMA vs. 1.01+/− 1.2 in healthy controls, *p* = 0.0113.

## 4. Discussion

We investigated the expression of some biomarkers of Th2 inflammation cytokines in infants with mild to moderate non-IgE-mediated CMA that needed to be fed with a hydrolyzed casein formula and showed also increased faecal calprotectin values.

In newborns, gastrointestinal symptoms are very common and suggest the presence of disorders such as food intolerance, infections, metabolic disorders, congenital disorders of the gastrointestinal tract, and, more rarely, allergies [8,13].

In pediatric clinical practice, it occurs frequently to assess infants who develop symptoms suggestive of allergy after being introduced to standard cow’s milk formula. If these symptoms appear after the intake of cow’s formula milk, food intolerance or allergy to cow’s milk proteins (CMPA) must be suspected and maybe IgE or non-IgE-mediated [14].

This clinical presentation is empirically treated by eliminating cow’s milk protein from the diet and special formula alternatives include hydrolyzed cow’s milk formula and amino acid-based formula that are all nutritionally adequate replacements of cow’s milk formula [6,8,9,14].

Nevertheless, the non-IgE-mediated CMA presents in the first period of life with a variety of symptoms, and in newborns, the most frequent at onset are the gastrointestinal ones: vomiting, diarrhea, constipation, severe colic, abdominal distension, and hematochezia. However, these manifestations are not specific to making the differential diagnosis, delaying the identification of these infants [6,13].

Moreover, in the field of neonatal medicine, an oral provocation (double-blinded, placebo-controlled) challenge test is not feasible because it can induce severe reactions and there are also no standardized diagnostic criteria. In common clinical practice, the specific IgE dosage and the execution of the Prik test in the newborn are also rarely performed, due to the poor reliability of the first examination and difficulty in carrying out the second, since newborns have intrinsic skin reactivity linked to age [13].

Thus, it follows that the diagnostic process described by the DRACMA guidelines (Diagnosis and Rationale for Action Against Cow’s Milk Allergy) [2] is difficult to apply in the neonatal setting [2,14].

Interestingly, in our study in the group of patients with mild to moderate non-IgE-mediated CMA when compared to peer healthy controls, increased mean values of IL-4 and IL-13 and decreased mean values of IL-5 and IL-10 were found.

The Th2 polarization is characterized by IL-4/IL-13 dependent pathway activation in atopic eczema, allergic asthma, and food allergies [15,16] and consistently we observed similar and significantly higher values of these cytokines. 

Interleukin-5 (IL-5) is predominantly regarded as a Th2–cytokine, which contributes to an allergic response by activating eosinophils [17]. Our finding showed lower values of IL-5 and this may be associated with our special category of non-IgE-mediated food allergy. 

Regulatory T cells are able to secret IL-10, a cytokine with inhibitory properties; in fact, it inhibits Th1 and Th2 cells’ activity as well as the function of the innate immune system. 

When compared with adult T cells, the producing capacity and receptor expression of neonatal T cells is known to be reduced [18]. Therefore, the findings of lower values of IL-10 have a theoretical background in the fact that the scarce maturation of IL-10 synthesis may be involved in the breaking of tolerance, characterizing the onset of atopic disease, as recently reported by Georgountzou et al. [15].

We have to consider that little is known about the immunologic maturation and cytokines pathways in the early stages of life in atopic subjects [16,17] and new research could lead to new targeted immunotherapies, such as interleukins inhibitors [19].

Activated cytokines, including also TNF-α, IFN-γ, IL-5, IL-8, and IL-10, play an important role in allergic disorders and may also affect immunologic development because cytokines levels change remarkably during the first year of life and are also influenced by some birth-related factors [17,18].

Our results showed significantly lower values of IL-5 and IL-10 in non-IgE-mediated food allergy subjects treated with an hydrolyzed milk formula: in this field, it is interesting to notice that it has been recently reported in an animal model that a probiotic may be useful to enhance the intestinal barrier and to reduce inflammation increasing IL-10 values [20].

On the other hand, recent studies found that fecal markers of intestinal inflammation, such as fecal calprotectin, are possibly markers of intestinal barrier permeability, intestinal inflammation, and barrier disfunction; in fact, this is confirmed in our study, as infants with non-IgE-mediated food allergy also showed increased values of fecal calprotectin compared to those of healthy subjects [21].

### Strengths and Limitations of the Study

This is the first study that aims to detect the expression of Th2 cytokines (IL-4, IL-5, IL-10, and IL-13) in infants with non-IgE food allergy and to compare it to peer healthy subjects. Interestingly we found that there were higher values of expression of IL-4 and IL-13, whereas values of IL-5 and IL-10 were lower. 

The study has some limitations. The most important one is the relatively small sample size in the non-IgE-mediated food allergy group, which may make it difficult to determine if our findings are conclusive. However, our study could lead to further investigations with larger cohorts of infants.

Secondly, we did not investigate the expression of Th2 cytokines (IL-4, IL-5, IL-10, and IL-13) before switching the formula to a hydrolyzed milk formula.

Considering the very young age of the enrolled infants, blood sampling should be limited as much as possible: however, in order to confirm the observed data in peripheral blood, further longitudinal studies should be specifically designed to assess cytokines expression over time before and after switching the formula.

## 5. Conclusions

Our data revealed that values of IL-4 and IL-13 are higher in infants with non-IgE-mediated food allergies, whereas IL-5 and IL-10 are lower, as observed in IgE-mediated food allergies [2]. The practical implication needs to be better clarified and further studies are needed to define the most effective dietary approach [22].

Nevertheless, our cytokines findings showed that an inflammatory response is still present after a few days from a special diet introduction.

This opens the way for future research with the aim of quantifying the cytokines response after a prolonged diet and better characterizing the causes of the onset of allergic symptoms in non-IgE-mediated food allergies in order to find some specific factors that could improve these conditions (prebiotics, probiotics).

## Figures and Tables

**Figure 1 nutrients-14-01565-f001:**
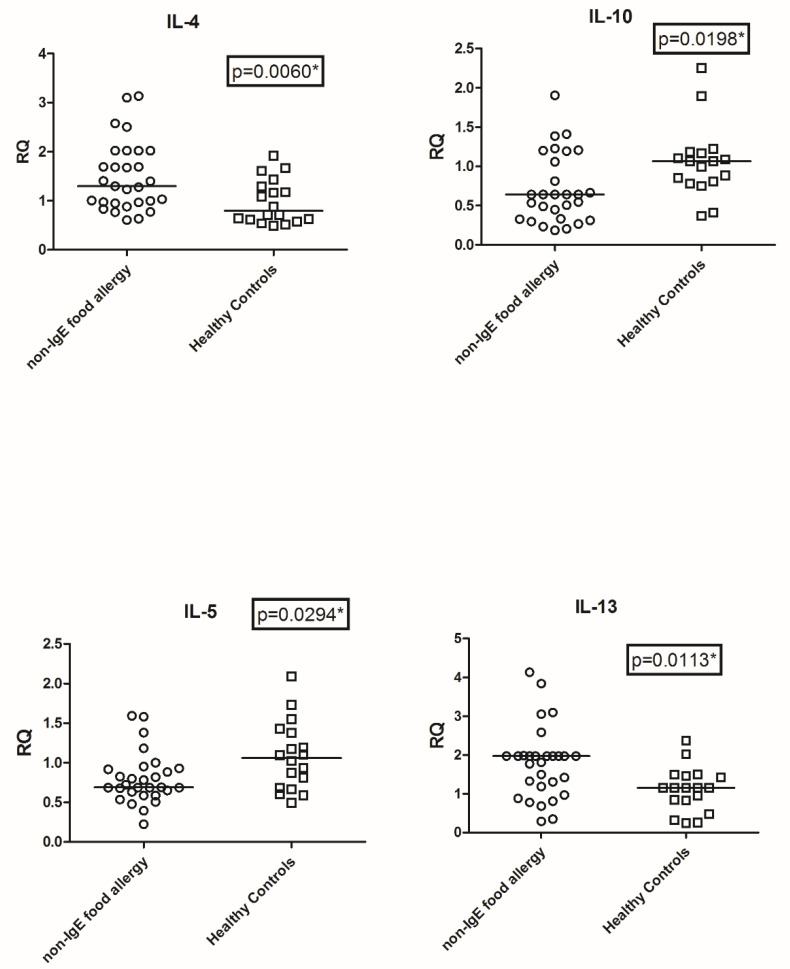
Transcription levels of IL4, IL5, IL10, and IL13 in Non-IgE-mediated food allergy and healthy controls. Footnote: Data are represented as box plots or circles. Relative ILs levels were assessed by real-time PCR and represented by fold change of Arbitrary Unity. RQ: Relative Quantitative units of fold change. Statistical significance was calculated by the Mann-Whitney *t*-test. Statistical significance has been set at *p* values of < 0.05, indicated with *.

**Table 1 nutrients-14-01565-t001:** Primer and probe.

Target	Sequence 5′→3′
GAPDHF	-CCAAGGTCATCCATGACAAC-
GAPDHR	-GTGGCAGTGATGGCATGGAC-
GAPDH-6FAM	-6FAM-TGGAGAAGGCTGGGGCTCAT-TAMRA
IL4F	-ACTTTGAACAGCCTCACAGAG-
IL4R	-TTGGAGGCAGCAAAGATGTC-
IL4P-6FAM	6FAM-CTGTGCACCGAGTTGACCGTA-TAMRA
IL5F	-GCTCTTGGAGCTGCCTACGT-
IL5R	-CAAGGTCTCTTTCACCAATGCA-
IL5P-6FAM	-6FAM-ATGCCATCCCCACAGAAATTCCCAC-TAMRA
IL10F	-ATGAAGGATCAGCTGGACAACTT-
IL10R	-CCTTGATGTCTGGGTCTTGGT-
IL10P-6FAM	-6FAM-ACCTGGGTTGCCAAGCCTTGTCTG-TAMRA
IL13F	-CTCATTGAGGAGCTGGTCAACA-
IL13R	-TCCATACCATGCTGCCATTG-
IL13P	6FAM-CACCCAGAACCAGAAGGCTCCGC-TAMRA

**Table 2 nutrients-14-01565-t002:** Subject characteristics.

	Non-IgE Food Allergy (n = 34)	Healthy Controls (n = 19)	*p*-Value
Age at enrollment			
Weeks (median, range)	10 (4–6)	8 (2–16)	0.248 *
Gender			
Female, n (%)	16 (47.0)	10 (52.6)	1 ^#^
Male, n (%)	18 (53.0)	9 (47.4)	1 ^#^
Birth weight			
Grams (median, range)	3180 (2960–3390)	3070 (2895–3235)	0.307 *
Gestational age			
Weeks (median, range)	36 (35–37)	37 (35–39)	0.250 *
Nationality			
Italian, n (%)	25 (73.5)	14 (73.7)	0.69 ^#^
Foreign, n (%)	9 (26.5)	5 (26.3)	0.693 *
Dietary treatment			
Amino acid formulas, n (%)	5 (14.7)	-	-
Hydrolyzed casein formulas, n (%)	29 (85.3)	-	-
Breastmilk, n (%)		12 (63)	
Standard formula, n (%)		7 (37)	

^#^ Fisher’s test, * Mann–Whitney *t*-test (*p* was set at 0.005).

**Table 3 nutrients-14-01565-t003:** Fecal calprotectin value.

	Non-Ig E Food Allergy (n = 34)	Healthy Controls (n = 19)	*p* Value
Fecal calprotectin			
µg/g (median, range)	2176 (366–4210)	368 (120–840)	0.015 *

* Mann-Whitney *t*-test (*p* was set at 0.005).

## Data Availability

Data are available in archive of our Department.

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
