# Peer review of "Analysis of Serum Th2 Cytokines in Infants with Non-IgE Mediated Food Allergy Compared to Healthy Infants"

_nutrients, 2022, doi:10.3390/nu14081565_

Round 1

Reviewer 1 Report

Non-IgE-dependent CMA most often affects the youngest children, and gastrointestinal symptoms are the main manifestation of the allergic process. For this type of allergy, the results of tests for the determination of specific asIgE antibodies against cow's milk proteins are negative. This is the reason why the participation of the allergic process in the development of clinical symptoms in these patients is not taken into account. This misunderstanding occurs commonly during the diagnostic process being conducted by general practitioners, gastroenterologists, and even allergists. The authors should be congratulated on their understanding of the pathogenetic mechanism of this type of allergy and on the choice of the PCR method to identify markers of allergic inflammation (cytokines, IL-4, IL-5, IL-10, IL-13) caused by stimulation of Th2 lymphocytes by cow's milk protein allergens. The results of the studies showed an increased level of interleukin 4 and interleukin 13 in the blood of patients. This proves that the allergic inflammatory process takes place in the organs of these children at the level of the cells of the digestive tract. An additional confirmation of the presence of an active inflammatory process is the increased level of calprotectin in the stools of the studied children (the second marker of the inflammatory process). I fully agree with the authors' considerations contained in the introduction and discussion. This publication brings new original scientific and practical value.

Author Response

We are gratefull for your kind comments.

Moreover we worked at our manuscript  to improve it and in order to satisfy the other reviewer's requests.

Reviewer 2 Report

The authors present the investigation of cytokines in serum samples of infants fed with formulas in a group with non-IgE-CMA or healthy control.

  • Most important point: in abstract and methods, it is not clearly described which children where included:
    • non-IgE-CMA already on formula diet? HA diet? AA diet?
    • what with the healthy group - also on formula diet?
    • at which time point of their age and/or disease and/or formula feeding durance have serum samples been taken?
    • My suggestion is to draw a clear scheme with time line and characteristics of children (healthy vs. non-IgE-CMA)
    •  
  • Title should be more concise, skip list of cytokines, explain that you analyzed serum samples
  • Is the formula feeding an important characteristic in your study group? otherwise delete from title
  • Introduction is really confused: paragraphs switch between explanation of immunologic profile of atopy/allergy to description of non-IgE-CMA and back again (several times)
  • explain briefly calprotectin as biomarker in intro
  • Did also test cytokines on protein level? If not, state this and explain why (e.g. low sample volume in young infants)
  • line 156: typos in IFN-listing?
  • line 211: IgE negative = important, put already as line 190
  • Table 2: info on diet of healthy control is missing/important
  • lien 217: insert units (RQ to GapDH....etc to numbers of results); also in Fig. 1 (y-axes)
  • line 227 "depicting mild..." what? (missing word?); RQ = ?
  • line 259, Ref 15 seems to be wrong here, line 260 Ref 13 incorrect, line 265 Ref 14/15 incorrect?
  • lines 281-284 move up to line 271
  • lower IL-5 needs to be discussed, as not going along IL-4, still Th2 cytokine
  • Conclusion: "these findings could support the indication of the dietary treatment of these subjects..." - why can you draw this conclusion?  explain better!
  • Lot of typos in the whole MS

Author Response

Manuscript ID: nutrients-1627838
Type of manuscript: Article
Title: Analysis of cytokines IL-4, IL-5, IL-10, IL-13 in infants fed with
hydrolyzed formulas: a comparison between infants with non-IgE-dependent food
allergy and healthy infants.
Authors: Francesco Savino *, Ilaria Galliano, Francesca Giuliani, Stefano
Giraudi, Paola Montanari, Valentina Daprà, Massimiliano Bergallo

ANSWER For REVIEWERS

The authors present the investigation of cytokines in serum samples of infants fed with formulas in a group with non-IgE-CMA or healthy control.

  • Most important point: in abstract and methods, it is not clearly described which children where included:
    • non-IgE-CMA already on formula diet? HA diet? AA diet?
    • what with the healthy group - also on formula diet?
    • at which time point of their age and/or disease and/or formula feeding durance have serum samples been taken?
    • My suggestion is to draw a clear scheme with time line and characteristics of children (healthy vs. non-IgE-CMA)

We agree with the comments , and requested details are now included in abstract and in methods section.

We have added the following sentences : “Controls were fed either breastmilk or standard formula. Blood samples were taken within one week of  special diet for cases.”

  • Title should be more concise, skip list of cytokines, explain that you analyzed serum samples

According to your request the title has been shortened

Now the title is is : Analysis of serum Th2 cytokines in infants with non-IgE-dependent food allergy compared to healthy infants.

  • Is the formula feeding an important characteristic in your study group? otherwise delete from title

Done

  • Introduction is really confused: paragraphs switch between explanation of immunologic profile of atopy/allergy to description of non-IgE-CMA and back again (several times)

We have re ordered the introduction, and moved some paragraphs

  • explain briefly calprotectin as biomarker in intro

Done

  • Did also test cytokines on protein level? If not, state this and explain why (e.g. low sample volume in young infants)

We have measured mRNA of cytokines and not proteins , since the methods is able to detect m RNA and this required smaller blood samples. This is very important for very young infants

  • line 156: typos in IFN-listing?

corrected

  • line 211: IgE negative = important, put already as line 190

We agree and moved the sentence.

  • Table 2: info on diet of healthy control is missing/important

We have inserted these informations in the main text.

  • lien 217: insert units (RQ to GapDH....etc to numbers of results); also in Fig. 1 (y-axes)

Glyceraldhyde tryphosphate dehydrogenase GapDH; RQ Relative Quantitative  units of fold change

  • line 227 "depicting mild..." what? (missing word?); RQ = ?

 We apologize for this mistake, now  we have correct it

  • line 259, Ref 15 seems to be wrong here, line 260 Ref 13 incorrect, line 265 Ref 14/15 incorrect?

We apologize for these unintentional mistakes, now we have corrected the references

  • lines 281-284 move up to line 271

Done, we have moved the paragraph

  • lower IL-5 needs to be discussed, as not going along IL-4, still Th2 cytokine

We have added this sentence in the Discussion section : “ However our finding showed a lower values for IL-5 , and it maybe associated with our special category of non Ig E mediated food allergy. “

  • Conclusion: "these findings could support the indication of the dietary treatment of these subjects..." - why can you draw this conclusion?  explain better!

According to you requests we have added new sentences

  • Lot of typos in the whole MS

 The whole manuscript has been revised foe an editing, and in particular Methods sections has been rewritten in order to remove all the repetitions and references has been updated

Round 2

Reviewer 2 Report

Most point addressed, but there are still some important issues:

* Table 2 urgently needs to contain the diet of healthy controls (percent formula vs. breastfeeding)

  • Conclusion: as you have had all study subjects on formula, you cannot conclude that formula-diet does confer any advantage. Skip the sentence "The practical implication need to be better clarified, but the improvement of symptoms could support the indication of the dietary treatment of these subjects with special hydrolyzed formulas as intended for allergic infants. " from line 523-327.
  • * Figure 1: indicate stat. sign. differenes directly in diagrams (asterisks, bars)

Author Response

Dear Prof. Zuccotti,

Editor

Nutrients,

Thank you for your decision allowing us to make a revision of our Manuscript entitled entitled  “Analysis of cytokines IL-4, IL-5, IL-10, IL-13 in infants fed with  hydrolyzed formulas: a comparison between infants with non-IgE-dependent food  allergy and healthy infants.”. Manuscript ID: nutrients-1627838

Authors: Francesco Savino *,  Francesca Giuliani, Stefano Giraudi, Ilaria Galliano, Paola Montanari, Valentina Daprà, Massimiliano Bergallo  for publication in Nutrients .

According the request of Reviewer 2,  we have modified also the  title as follow: Analysis of serum Th2 cytokines in infants with non-IgE-dependent food allergy compared to healthy infants.

Now all the issues raised by the reviewer have addressed.

Responses to the referees’  comments.

 SECOND REVISION

Most point addressed, but there are still some important issues:

* Table 2 urgently needs to contain the diet of healthy controls (percent formula vs. breastfeeding)

Now we have added these details in the table 2.

  • Conclusion: as you have had all study subjects on formula, you cannot conclude that formula-diet does confer any advantage. Skip the sentence "The practical implication need to be better clarified, but the improvement of symptoms could support the indication of the dietary treatment of these subjects with special hydrolyzed formulas as intended for allergic infants. " from line 523-327.

According to your request we have deleted the sentence : “  but the improvement of symptoms could support the indication of the dietary treatment of these subjects with special hydrolyzed formulas as intended for allergic infants”   We have added  “ and further studies are need to define the most effective dietary approach.”

  • * Figure 1: indicate stat. sign. differenes directly in diagrams (asterisks, bars)

We have modified the Figure1 and statistical differences are now reported .

These data have not been published previously and are not under consideration for publication by any other journal. All authors have read and approved the version of the article being submitted.

I am looking forward to receive your comments and we hope in a favourable outcome and subsequent publication in Nutrients.

Round 3

Reviewer 2 Report

Manuscript sufficiently improved, some minor English typos

Author Response

-